# Fabrication of Superhydrophobic Composite Membranes with Honeycomb Porous Structure for Oil/Water Separation

Chunling Zhang [1], Yichen Yang [1], Shuai Luo [1], Chunzu Cheng [2,*], Shuli Wang [2] and Bo Liu [3,*]

1   School of Materials Science and Engineering, Jilin University, Changchun 130022, China
2   State Key Laboratory of Bio–Based Fiber Manufacturing Technology, China Textile Academy, Beijing 100025, China
3   Key Laboratory of Bionic Engineering, Ministry of Education, Jilin University, Changchun 130022, China
*   Correspondence: chunzuc@163.com (C.C.); lbb1107@jlu.edu.cn (B.L.)

**Abstract:** Due to the low separation efficiency and poor separation stability, traditional polymer filtration membranes are prone to be polluted and difficult to reuse in harsh environments. Herein, we reported a nanofibrous membrane with a honeycomb–like pore structure, which was prepared by electrospinning and electrospraying. During the electrospraying process, the addition of poly-dimethylsiloxane and fumed $SiO_2$ formed pores by electrostatic repulsion between ions, thereby increasing the membrane flux, subsequently reducing the surface energy, and increasing the surface roughness. The results show that when the content of $SiO_2$ reaches 1.5 wt%, an ultra–high hydrophobic angle ($162.1° \pm 0.7°$) was reached. After 10 cycles of oil–water separation tests of the composite membrane, the oil–water separation flux and separation efficiency was still as high as 5400 L m$^{-2}$ h$^{-1}$ and 99.4%, and the membrane maintained excellent self–cleaning ability.

**Keywords:** superhydrophobic; oil/water separation; electrospraying; electrospinning

## 1. Introduction

A large amount of industrial oily wastewater was discharged everywhere, which has become a serious environmental problem and has endangered our environment [1–3]. Currently, how to treat polluted water efficiently and quickly has become a problem and challenge. Several common treatment methods such as the physical separation method, chemical separation method, biological separation method, and so forth were applied to address this issue [4–6]. However, these methods have complicated operations, make processes cumbersome, show a poor separation performance, and are prone to generating toxic gas and causing secondary pollution [7,8]. Hence, the membrane separation method has become a hot topic in both industry and academia because of its simple operation, low energy consumption, and high separation efficiency [9–12].

Hydrophobic materials are effective in treating oily sewage, therefore the preparation of superhydrophobic surfaces is the most critical step. A superhydrophobic surface is defined as a surface carrying a water contact angle greater than 150° and low contact angle hysteresis [13–15]. The preparation of superhydrophobic surfaces is in two basic principles: (1) Reduce the surface energy by changing the chemical composition of the compound. (2) Reduce the surface energy by changing the surface roughness. Accordingly, how to prepare a hydrophobic surface with low surface energy is the key to solving the problem [16–19].

Recently, PVDF has been widely used in the field of oil–water separation due to its superior mechanical strength and its hydrophobic groups [20–23]. Due to the hydrophobicity of the polyvinylidenefluoride (PVDF) membrane itself, it is easily contaminated when separating oil–in–water emulsions, which reduces the service life of the membrane and increases the cost of oil–water separation [23–25]. Endowing the superhydrophobicity of the film can improve the antifouling performance of the film. In recent years, the blending of inorganic nanoparticles in polymer membranes has attracted attention. It has been

demonstrated that the blending of the inorganic filler has led to an increase in membrane permeability and better control of membrane surface properties [26–28]. Nano $SiO_2$ is a new type of non–toxic, odorless, and pollution–free new ultrafine inorganic material that has a small particle size, high aspect ratio, large specific surface area, and good dispersibility [29]. The abundant hydroxyl groups on the surface enhance the surface effect, have good compatibility with the membrane material and produce a hierarchical structure, which greatly improves the flux and separation efficiency. Yang et al. [24] successfully prepared PVDF rough nanofiber composite membranes by one–step electrospinning technology, with a water contact angle of 135° and an oil–water separation efficiency of 93.9%. Gao et al. [30] used electrospinning and electrostatic spraying to prepare PVDF–$SiO_2$ nanofiber membranes with surface microsphere structure, the water contact angle reached 152°, and the oil–water separation efficiency was as high as 97%. Although their contact angles are quite high, their flux has a huge drop in 10 oil–water separation cycles, and drops from 7000 to 4000 L $m^{-2}$ $h^{-1}$ after 10 oil–water separation cycle experiments. The reason is that the surface morphology of the composite membrane is composed of electrospinning micropores and microspheres, which are easily polluted and blocked during the oil–water separation process. In order to solve the self–cleaning problem, researchers usually add hydrophobic groups to reduce the surface energy and modify the surface morphology of the membrane [31–34]. Generally, groups containing F and Si elements can reduce the surface energy of the film and increase the hydrophobicity of the film [35–37]. However, to prevent the secondary pollution caused by the F element in the oil–water separation process, a long section of siloxane to the membrane is preferred to achieve the superhydrophobic surface. Polydimethylsiloxane (PDMS) is a fluorine–free polymer. Due to its low surface energy and stable chemical properties, it is suitable for preparing various hydrophobic membranes [38–40]. In this study, we used the hydrosilylation reaction to introduce the rigid group with phenol into the polysiloxane to synthesize a new type of polydimethylsiloxane (DP8) with the long chain segments, and finally prepared a composite film by electrospinning and the electrospray technology with a honeycomb porous structure [41]. In combination with nano $SiO_2$ and siloxane segments together for modifying the PVDF, superhydrophobicity and self–cleaning were achieved. The oil–water separation performance, stability, and reuse rate of PVDF/DP8/$SiO_2$ composite membrane for oils of different densities were investigated.

## 2. Experiments

### 2.1. Materials

PVDF was purchased from National group chemical reagent Co., Ltd. (Beijing, China). 1,1,3,3–Tetramethyldisiloxane (TMDS) and octamethyl cyclotetrasiloxane (D4) were purchased from Aladdin Technology Co., Ltd (Shanghai, China). The DP8 was synthesized according to our published protocol [41]. Hydrophobic $SiO_2$ were purchased from Aladdin Reagent Co., Ltd. (Shanghai, China). Methyl blue stain, Sudan III stain, 4,4'-dihydroxydiphenyl, $K_2CO_3$, bromopropane, methylene chloride, n-hexane, chloroform, carbon tetrachloride, petroleum ether, N,N-dimethylformamide (DMF), tetrahydrofuran (THF), acetone, anhydrous ethanol were obtained from Beijing Chemical Industry (Beijing, China).

### 2.2. Preparation of Membranes

Synthesis of DP8 is through the silyl–hydrogen reaction of TDMS and D4 to form 2H–PDMS–10, while 4,4'–dihydroxydiphenyl, $K_2CO_3$ and bromopropene generate intermediate products, then through Claisen rearrangement reaction to generate 3,3'–diallyl–biphenyl–4,4'–diol(DABP), and finally through polymerization reaction to generate long chain polymer, DP8 (Scheme 1) [41]. The PVDF/DP8/$SiO_2$ membranes were fabricated via simple electrospinning and electrospray technology. Then, 1 g PVDF was dissolved in 10 mL of mixed solvents of DMF and acetone ($v/v$ = 2:3), the blend solutions were subsequently electrospun at a feeding rate of 0.5 mL $h^{-1}$ with 16 kV applied voltage between the working and collecting electrode (receiving distance was set to 12 cm). The synthesized

PVDF nanofiber membrane was placed in a vacuum oven and dried for 24 h, as shown in Scheme 2.

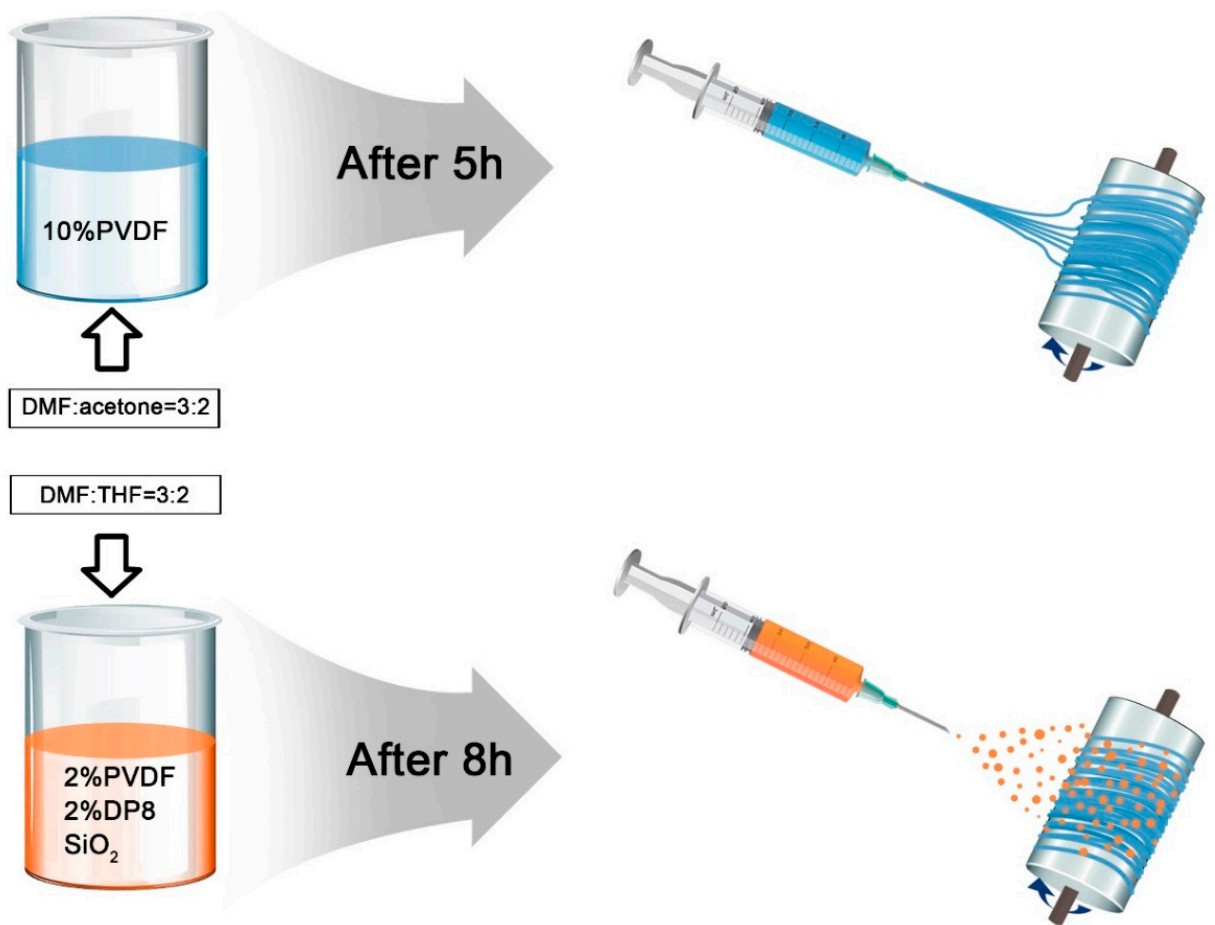

**Scheme 1.** Synthesis of DP8.

**Scheme 2.** Electrospinning and electrospraying schematic diagram.

The different weight proportions of $SiO_2$ (0, 0.5%, 1.0%, 1.5%, 2.5%, 4%), 2 wt% PVDF and 2 wt% DP8, were dissolved in mixed solvents of DMF and THF (*v:v* = 6:4) and stirred at room temperature until completely dissolved, and then the membrane was fabricated by electrospinning. Here are the parameters of electrospinning: applied voltage of 18 kV, receiving distance of 12 cm, feeding rate of 0.4 mL/h, temperature 20 °C, humidity below 30%. Finally, we took out the double–layer membranes and dried them at 50 °C for 24 h. The obtained membrane was noted as DP8–X, where X presents the concentration of $SiO_2$ in a mixed solution.

### 2.3. Characterizations

The surface chemical structure of DP8–X composite membrane was analyzed by Fourier transform infrared (FT–IR) spectra ranging from 4000 to 400 cm$^{-1}$ with a FT–IR system (Nexus 670. Nicolet, WI, USA). Observing the surface morphology of the PVDF/DPn composite membrane with scanning electron microscopy (SEM, FEI XI,30 ESEM FEG). The surface of the sample was sprayed with gold for three minutes, in order to increase the conductivity of the sample. The thermal stability of PVDF/DPn/SiO$_2$ composite membrane analysis was evaluated by Differential Scanning Calorimeter (DSC) (Q20 thermal analyzer, TA, New Castle, PA, USA) and Thermogravimetric measurement (TGA, Perkin–Elmer, Waltham, MA, USA). DSC and TGA experiments were carried out under nitrogen protection, heated with about 3–5 mg DSC sample from 60 to 240 °C at a rate of 10 °C min$^{-1}$. TGA was heated from 30 to 800 °C (10 °C min$^{-1}$). Obtained water contact angles were analyzed with a Drop Shape Analyzer (DSA100, Kruss, Hamburg, Germany) by dropping deionized water perpendicularly with a 2 μL syringe onto the PVDF/DP8/SiO$_2$ membrane surface at room temperature. Each membrane was measured five times and the average value recorded.

### 2.4. Oil–Water Separation

To measure the oil–water separation performance of PVDF/DP8/SiO$_2$ membranes, the PVDF/PD8/SiO$_2$ membrane was placed in the separation device, and the separation device was placed vertically. The effective separation area of the membrane was calculated to be 4.41 cm$^2$. We poured 60 mL of a mixture of dichloromethane and water ($v$:$v$ = 1:1) into the glass container above and used gravity as the driving force to separate oil and water. To ensure the separation was complete, the system was maintained for 5–10 min, and two barrels were used to collect oil and water. The equation of separation efficiency was as follows [42,43]:

$$\Phi = V_1/V_2 \tag{1}$$

where $\Phi$ is the separation efficiency, and $V_1$ and $V_2$ are the volume of oil before and after separation (mL), respectively. The equation of the oil–water separation flux was as follows [44,45]:

$$Flux = V/A_t \tag{2}$$

where V is the volume of oil phase passing through the membrane (L), $A_t$ is the effective area of separation membrane (m$^2$), and t is the separation time (h).

## 3. Results and Discussions

### 3.1. Preparation of PVDF/DP8/SiO$_2$ Composites

After adding DP8 and SiO$_2$ to the electrospray solution, DP8 and PVDF were randomly distributed in all parts of the PVDF fiber. Figure 1 shows the FTIR spectra of pure PVDF nano-fiber membrane, PVDF/DP8 composite membrane, and the PVDF/DP8/SiO$_2$ nanofiber membranes with different content additives (DP8–X). The PVDF nanofiber membranes show obvious C–H vibration peaks and C–F vibration peaks at 1400 and 1168 cm$^{-1}$, respectively. When the copolymer DP8 was added, the –CH$_3$ and Si–CH$_3$ bend vibration peaks and Si–O–Si stretching vibration peaks at 2963, 1280, and 1168 cm$^{-1}$ are observed, which indicates that DP8 was successfully added to the PVDF fiber. In the membrane, when SiO$_2$ nanoparticles were added to PVDF/DP8, Si–O stretching vibration peaks became more obvious, and then –CH$_3$, Si–CH$_3$, and C–F bending vibration peaks gradually disappeared because the coated SiO$_2$ nanoparticles on the surface of PVDF/DP8 after the increasing content of SiO$_2$ weakened the intensity of these peaks. Furthermore, this also confirms that SiO$_2$ nanoparticles were successfully doped into PVDF/DP8.

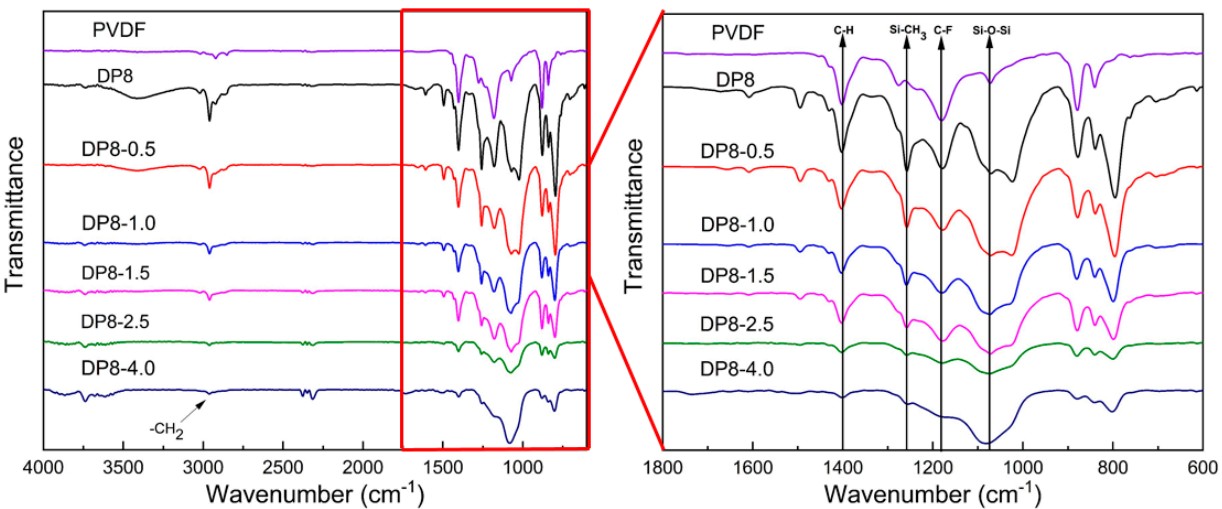

**Figure 1.** FTIR of pure PVDF nanofiber membrane and DP8–X fiber membrane.

The represented element distribution of PVDF/DP8/SiO$_2$ composite film (DP8–1.5) was analyzed by EDS (Figure 2). The C, O, and F elements were evenly distributed on the surface of the film, and the distribution of Si had a slight reunion. This is because with the increase in the SiO$_2$ content, the phenomenon of reunion between inorganic ions and polymers, and the performance of the membrane are also affected by this phenomenon [30].

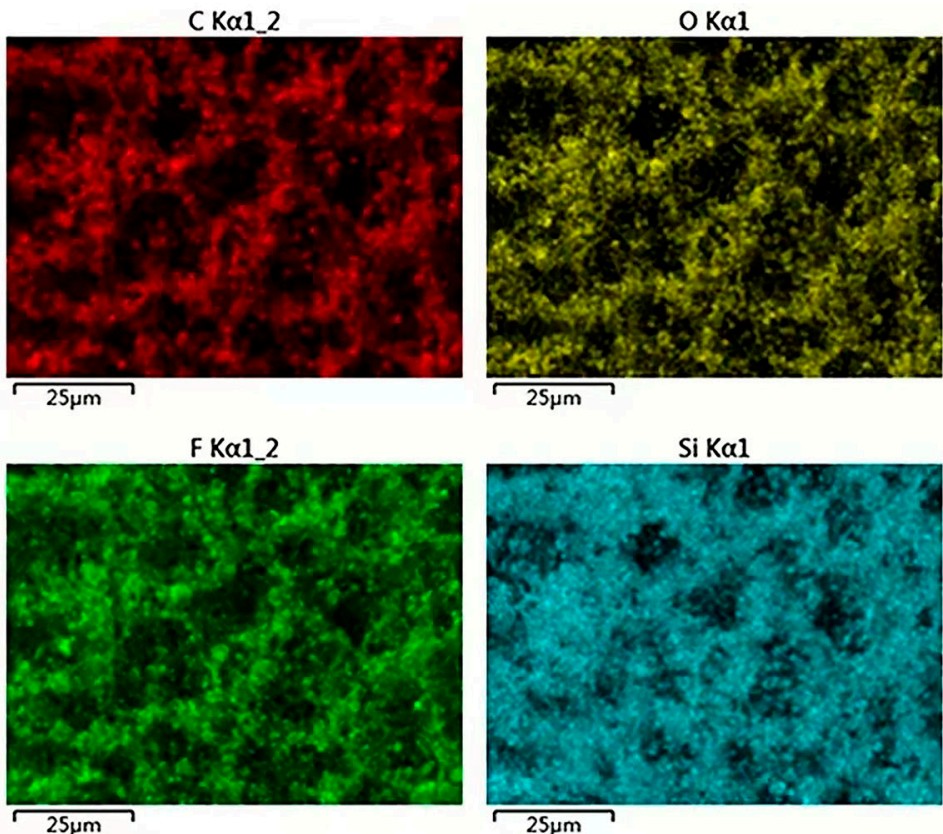

**Figure 2.** EDS analysis of DP8–1.5 layered membrane: C, O, F, Si element distribution.

*3.2. Thermal Performance Analysis of PVDF/DP8/SiO$_2$ Composite Membranes*

The thermal stability of the composites was studied by thermogravimetric analysis [46–48]. Thermal stability has always been the key to long–term stable use of oil–water

separation membranes [49–51]. Figure 3a shows the thermal weight loss curve of pure PVDF, PVDF/DP8, and DP8–X composite nanofiber membranes. The thermal decomposition process of all membranes was completed in one step, with decomposition beginning at 393 °C and reaching Tmax at 457 °C, and the final Char residues at 800 were 23%. Figure 3b shows the DSC curves of pure PVDF, PVDF/DP8, and DP8–X composite nanofiber membranes. All the films showed a broad endothermic peak at about 160 °C, which is the melting point of the film. After adding SiO$_2$ nanoparticles, all the endothermic peaks did not move significantly, indicating that the addition of SiO$_2$ nanoparticles did not reduce the melting point of pure PVDF. Taking the TGA and DSC results together, the addition of SiO$_2$ nanoparticles and siloxane chain segments did not affect the thermal stability of the membranes.

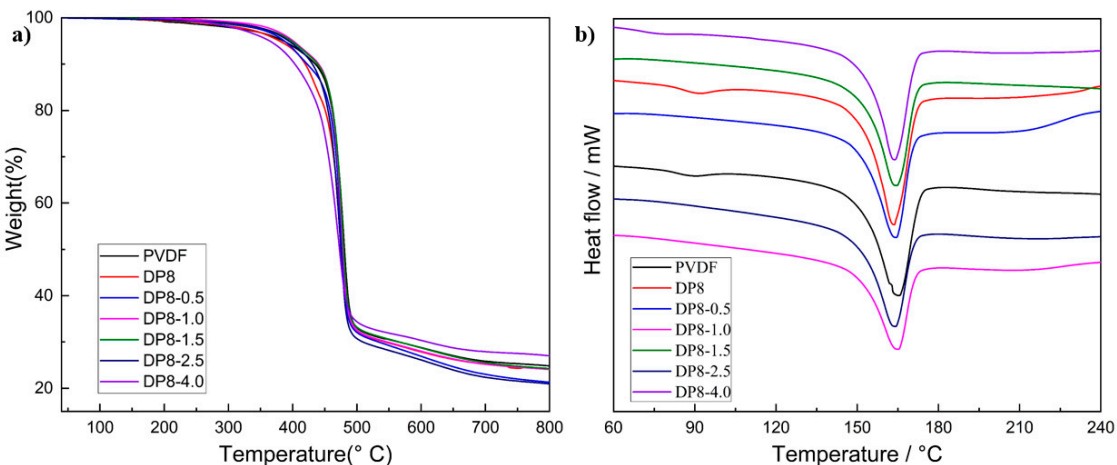

**Figure 3.** (**a**) TGA and (**b**) DSC curves of pure PVDF nanofiber membrane and DP8–X fiber membranes.

### 3.3. PVDF/DP8/SiO$_2$ Composite Membrane Water Contact Angle Test and Self–Cleaning Ability Test

Hydrophobicity is an important parameter of oil–water separation performance [52,53]. The water contact angle is tested under dry equilibrium conditions. Figure 4 shows the water contact angles of pure PVDF, PVDF/DP8, and DP8–X composite nanofiber membranes. The water contact angle of the pure PVDF nanofiber membrane was only 121.9° ± 0.7°. However, when the electrospray process fabricated the PVDF/DP8 microspheres on the PVDF nanofiber substrate, the hydrophobic angle of the separation membrane increased from 121.9° ± 0.7° to 145.3° ± 0.8°. This is mainly attributed to the addition of Si-CH$_3$ to reduce its surface energy, and the microsphere structure to increase the surface roughness [36,41], thus improving the hydrophobicity. The hydrophobic angles of DP8–0.5, DP8–1.0, DP8–1.5, and DP8–4.0 were 159.3° ± 2.3°, 159.5° ± 1.7°, 162.1° ± 0.7°, 153.6° ± 2.6°, and 150.4° ± 1.8°, respectively. The hydrophobic angles of the membranes were significantly increased and all the angles reached the values of superhydrophobicity, after incorporating the hydrophobic gas phase SiO$_2$. When the amount of hydrophobic gas phase SiO$_2$ nanoparticles reached up to 4%, the hydrophobic angle was significantly reduced. This is because the interface compatibility between the hydrophobic gas phase SiO$_2$ nanoparticles and PVDF becomes worse, and the agglomeration forms on the surface when the content of SiO$_2$ nanoparticles is too high [43].

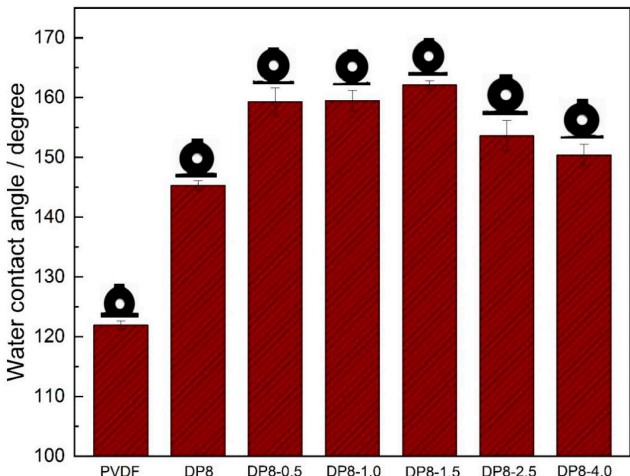

**Figure 4.** WCAs of pure PVDF nanofiber membrane and DP8–X fiber membranes.

*3.4. Oil–Water Separation Test of PVDF/DP8/SiO₂ Composite Membrane and Self–Cleaning Ability Test*

According to the above analysis, the PVDF/DP8/SiO$_2$ membrane showed excellent hydrophobicity. We used the DP8–1.5 composite membrane for the oil–water separation experiment to further evaluate the oil–water separation performance of the membrane. As shown in Figure 5, 30 mL of water (methyl blue stain) and 30 mL of dichloromethane (Sudan III stain) were configured to form a 60 mL oil–water mixture and placed in a beaker. The DP8–1.5 composite membrane was fixed in the middle of the separation device, and the mixture was slowly poured into the upper funnel. When the oil–water mixture contacted the DP8–1.5 composite membrane, the oil–water mixture was selectively passed through due to the hydrophobic and lipophilic properties of the membrane. Dichloromethane quickly penetrated and passed through the DP8–1.5 superhydrophobic membrane, and finally the oil droplets were collected in the lower beaker. Without any external force, water did not pass through the DP8–1.5 composite membrane, and the separation process was rapid. In order to ensure that all oil droplets could pass through the DP8–1.5 composite membrane, the oil–water separation performance measurement was performed after the entire separation process was maintained for 10 min. Figure 6 shows that the honeycomb porous structure membrane exhibited excellent oil flux and oil–water separation efficiency in the oil–water separation process. The measured oil flux and oil–water separation efficiency could reach up to 5000 L m$^{-2}$ h$^{-1}$ and 99.95%. It is obvious that the flux and separation efficiency was the best when the SiO$_2$ concentration was 1.5%. As shown in Figure 6, the PVDF/DP8/SiO$_2$ composite membrane was used as the separation membrane to carry out 10 cycles of oil–water separation repeatability test. Then the membrane was soaked in absolute ethanol for 5 min and then cleaned and subjected to ultrasonic treatment for 10 min to test the repeatability. Through the above test, it was found that the hydrophobicity of the membrane was not reduced and the superhydrophobic state could be maintained. Within 10 cycles of testing, the oil fluxes of all PVDF/DP8/SiO$_2$ composite membranes remained above 4500 L m$^{-2}$ h$^{-1}$.

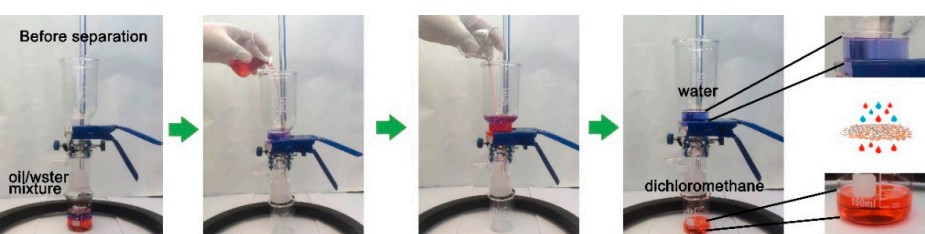

**Figure 5.** Dichloromethane–water mixture oil–water separation process diagram.

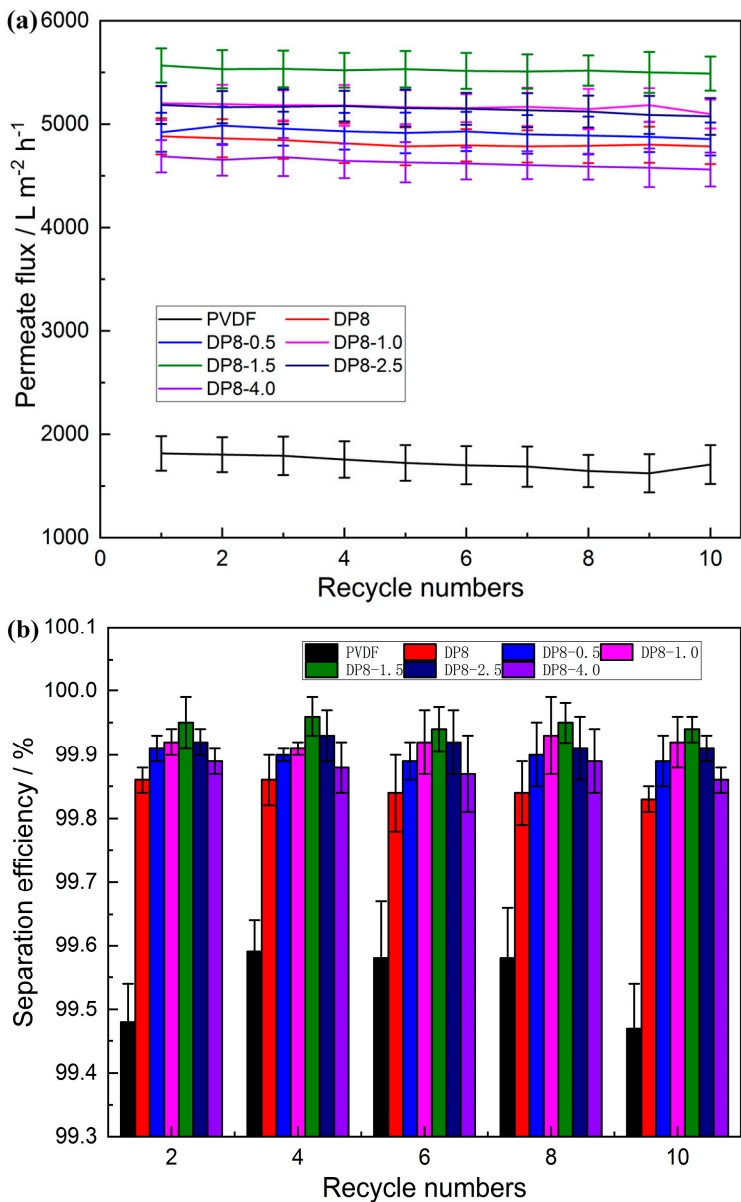

**Figure 6.** (**a**) Oil–water separation flux (**b**) efficiency of PVDF nanofiber membrane and DP–X layered membrane.

In order to study the oil–water separation performance of the DP8–1.5 superhydrophobic composite membrane more comprehensively, five different types of dichloromethane–water, n–hexane–water, chloroform–water, carbon tetrachloride–water and petroleum ether-water were applied for the oil–water mixing system and oil–water separation test. We used the same experimental method for separating the mixture, and the results are shown in Figure 7. When the density of oil was less than the density of water, we tilted the instrument at 45°, similar to n–hexane and petroleum ether. When five different mixtures were separated by the DP8–1.5 honeycomb porous structure membrane, it was found that all the oil fluxes and separation efficiencies were similar. The oil fluxes were 5552, 5498, 5511, 5489, 5545 L m$^{-2}$ h$^{-1}$, respectively. When the content of SiO$_2$ was 1.5%, it showed a high oil flux to the different oils. The separation efficiency for different oils could be maintained at more than 99.94%. It can be seen that the DP8–1.5 composite membrane had an excellent oil–water separation performance for different oil–water mixtures.

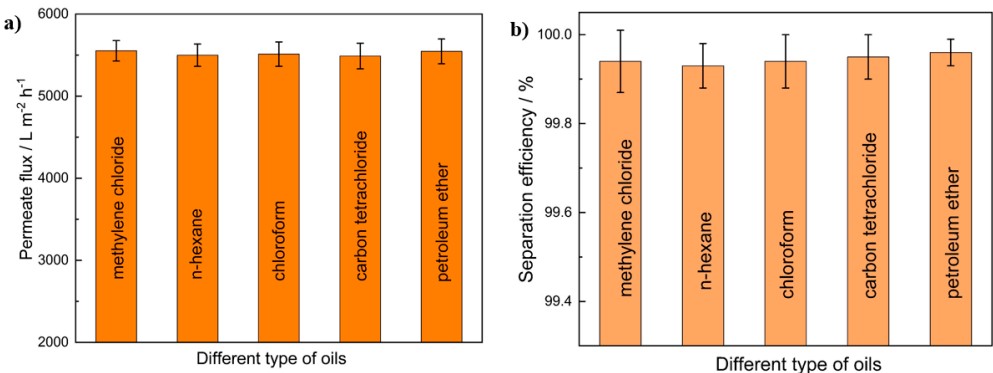

**Figure 7.** (**a**) Separation fluxes and (**b**) efficiencies of DP8–1.5 layered membrane for different oils.

### 3.5. Stability Test of PVDF/DP8/SiO$_2$ Composite Film

The long–term stability of the membrane is very important to the oil–water separation performance [54–57]. The membrane's water pollution resistance and self–cleaning ability were tested. In total, 4 μL deionized water was slowly and vertically dropped from the needle tube onto the DP8–1.5 composite membrane. When the water contacted the membrane surface in a large area, the water droplets were slowly lifted. Figure 8a clearly shows that the water droplets bounced off the surface, and the shape of the water droplets did not change during the entire experiment, did not fall off the needle under greater force, and did not adsorb to the surface, which shows that the modified film had a good anti–water adhesion and self–cleaning ability. As shown in Figure 8b, when the surface of the membrane was sprayed with water contaminated by methyl blue, the water rebounded and separated from the surface of the membrane, leaving no traces on the surface of the membrane. In addition, the methyl blue powder and sand grains were scattered on the surface of the membrane, and then the water was sprayed onto the surface of the membrane and rolled down the surface immediately. Moreover, the pollutants were taken away without leaving any traces and stains. In comparison, pure PVDF nanofiber membrane is easily contaminated under the same test [24,30].

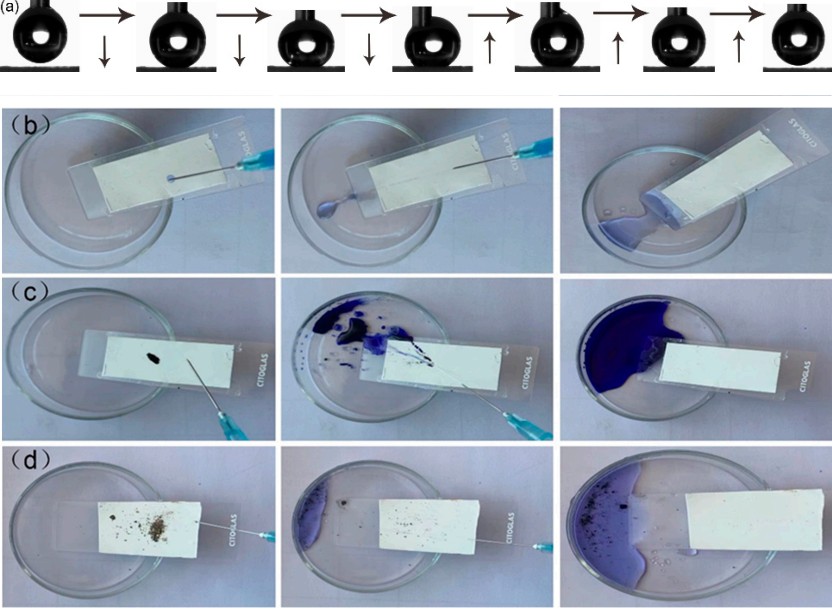

**Figure 8.** Self–cleaning ability of DP8–1.5 fiber membrane. (**a**) Water droplet drop experiment. (**b**) Water rebounds on the surface of an ultra–hydrophobic surface. (**c**) Methyl blue was removed by water droplets. (**d**) Sand grains were removed by water droplets.

To further study the stability of the membrane, the DP8–1.5 composite membrane was soaked in an acid–base salt solution for 24 h, and then the contact angle of the membrane was tested. Figure 9 shows the water contact angle (WCA) values immersed in the different solutions for different times. When immersed in an acid–base salt solution for 4, 8, 16, and 24 h, the contact angle of DP8–1.5 superhydrophobic composite film still showed superhydrophobicity. This shows that the DP8–1.5 layered structure membrane still showed excellent oil–water separation performance under different acid–base salt conditions.

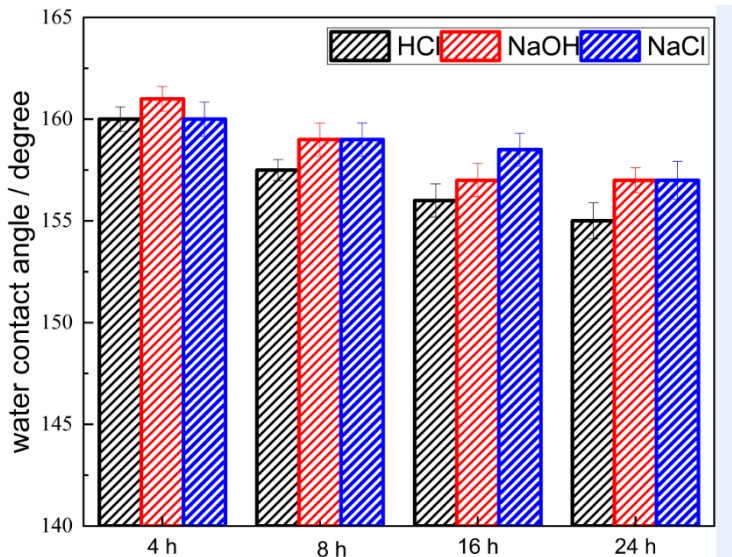

**Figure 9.** WCA test of DP8–1.5 layered membrane immersed in acid–base salt solution.

### 3.6. PVDF/DP8/SiO$_2$ Composite Film Surface Appearance and Mechanism Analysis

The structure of PVDF/DP8/SiO$_2$ was investigated by SEM. SEM images in Figure 10 showed the evolution of morphology of electrospun PVDF and electrospray DP8–X with different SiO$_2$ content. It is evident from a$_2$–g$_2$ of Figure 10 that the electrospun composite membrane had a distinct hierarchical structure. From Figure 10b$_1$–g$_1$, when SiO$_2$ and DP8 were added to the surface of the PVDF nanofiber matrix, microspheres with a hierarchical structure were obtained with the nanometer structure. It can be seen in Figure 10 b$_1$ that the surface of the microspheres formed by PVDF/DP8 is smooth. When SiO$_2$ was added, the surface of the microspheres was obviously wrinkled because SiO$_2$ was added to the microspheres. It can be seen from FTIR (Figure 1) that because DP8 itself ha phenolic hydroxyl groups, the phenolic hydroxyl groups themselves can be ionized. During electrospraying, the uncured microspheres formed an obvious honeycomb structure due to the interaction between the Coulomb force [41]. When the content of SiO$_2$ continued to increase to more than 2.5%, the surface honeycomb structure obviously disappeared (Figure 10b–e) which directly affected the performance of the membrane. As shown in WCA (Figure 6) test, with the increase in SiO$_2$ content, the oil flux increased significantly, mainly because the addition of SiO$_2$ can also form a good honeycomb structure, forming more oil channels. However, when the content of SiO$_2$ increased to 4%, the flux of the membrane was significantly reduced. When the higher content of SiO$_2$ was added, the hydrophobic gas phase SiO$_2$ nanoparticles coated the surface of the polymer and the phenolic hydroxyl functional groups of DP8 were covered. Hence, the uncured microspheres were not charged and could not generate Coulomb force, resulting in the disappearance of the honeycomb structure and reducing the oil channels and reducing the flux of oil–water separation during the electrospraying process.

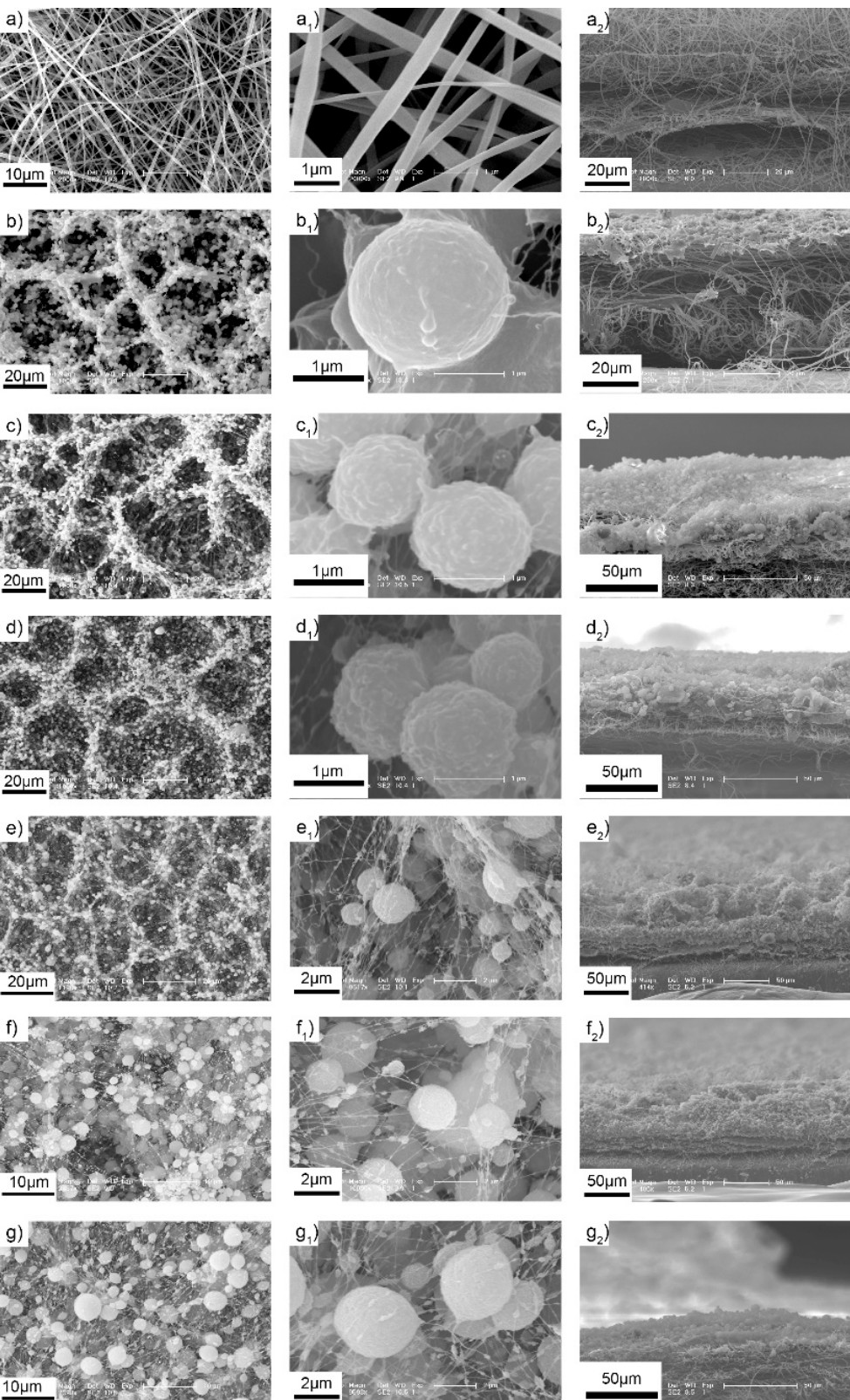

**Figure 10.** SEM of pure PVDF nanofiber membrane DP8 composite membrane and electrosprayed DP8–X with different SiO$_2$ content. (**a–a$_2$**) PVDF, (**b–b$_2$**) DP8, (**c–c$_2$**) DP8–0.5, (**d–d$_2$**) DP8–1.0, (**e–e$_2$**) DP8–1.5, (**f–f$_2$**) DP8–2.5, (**g–g$_2$**) DP8–4.0.

The hydrophobic model of the composite membrane is shown in Figure 11. PVDF was applied as the electrospinning substrate, and DP8 and SiO$_2$ were sprayed on the composite membrane by electrospraying. The Si–CH$_3$ bond in DP8 reduced its surface energy, and meanwhile, a honeycomb–like through–hole structure was formed due to the effect of electrostatic repulsion. This provided more channels and volume for conveying oil, thereby increasing oil flux. Moreover, the surface of the microspheres was coated with SiO$_2$ nanoparticles with a low surface energy, which increased the surface roughness of the membrane, and finally formed a composite membrane with a hierarchical structure and a honeycomb porous structure, which greatly enhanced the hydrophobicity.

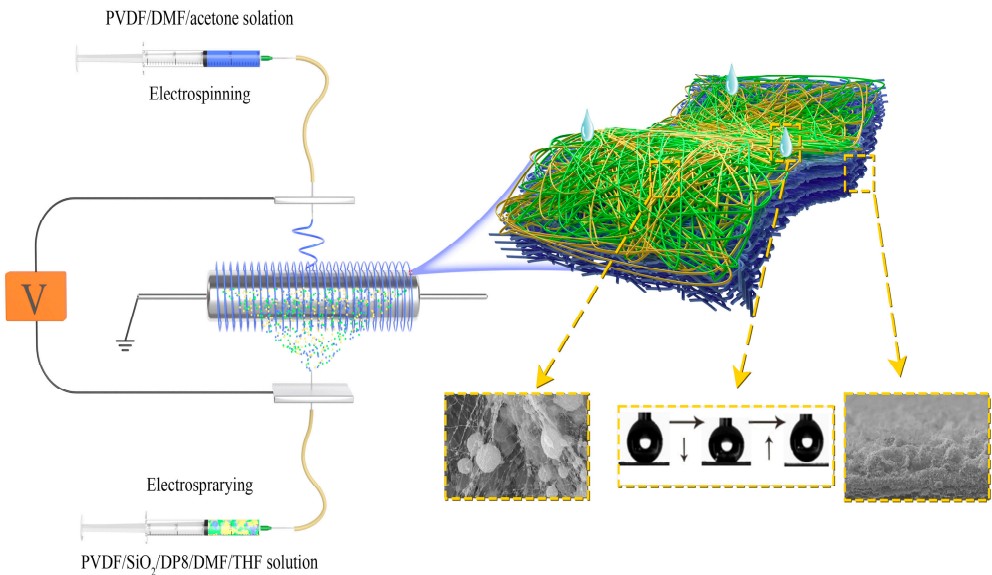

**Figure 11.** Hydrophobic model of PVDF/DP8/SiO$_2$ composite membrane.

*3.7. Conclusions*

This article mainly studies the preparation of PVDF/DP8/SiO$_2$ composite membranes with hierarchical structure and their performance in oil–water separation. During the electrospraying process, the addition of long segment polydimethylsiloxane and fumed SiO$_2$ formed pores through electrostatic repulsion between ions, thereby increasing the membrane flux, reducing its surface energy, and increasing its surface roughness. When the new polydimethylsiloxane DP8 and SiO$_2$ nanoparticles were added, the hydrophobicity of the film was greatly improved, and the hydrophobic WCA reached 162° ± 0.7°, which is much higher than pure PVDF. After 10 cycles of oil–water separation experiments, the separation efficiency of the composite membrane was still maintained above 99.94%. Because of the chemical composition, low surface energy, and honeycomb structure of the membrane, it has an ultra–high oil flux (>5400 L m$^{-2}$ h$^{-1}$), separation efficiency, and self–cleaning ability.

**Author Contributions:** Conceptualization, C.Z. and Y.Y.; methodology, S.L.; software, Y.Y.; validation, B.L., C.Z. and C.C.; formal analysis, Y.Y.; investigation, B.L.; resources, S.W.; data curation, Y.Y.; writing—original draft preparation, Y.Y.; writing—review and editing, Y.Y.; visualization, C.Z.; supervision, C.Z.; project administration, C.C.; funding acquisition, S.W. All authors have read and agreed to the published version of the manuscript.

**Funding:** We thank the funding support from department of science and technology of Jilin Province (20220201111GX) and open research fund of state key laboratory of bio–based fiber manufacturing technology, China textile academy (SKL202203).

**Institutional Review Board Statement:** Not applicable.

**Informed Consent Statement:** Not applicable.

**Data Availability Statement:** Not applicable.

**Conflicts of Interest:** The authors declare no conflict of interest.

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
