# Peer review of "Fabrication of Superhydrophobic Composite Membranes with Honeycomb Porous Structure for Oil/Water Separation"

_coatings, doi:10.3390/coatings12111698_

Round 1

Reviewer 1 Report

- The similarity index, investigated using turnitin software, is high (44 %) and should be reduced. In addition, there is a similarity of 6 % with the following reference:

https://www.sciencedirect.com/science/article/abs/pii/S0032386121001452?via%3Dihub

- The introduction doesn’t contain references published in 2022 which include the modified PVDF electrospun nanofiber membrane for oil water separation.

- The method used for oil water separation is only suitable for solvents with density higher than water such as halogenated solvents used where the oil layer is the bottom layer but it is not suitable for solvents with density lower than water such as hexane and petroleum ether. So, clarify this point and discuss the separation process for all hydrophobic solvents in the experimental work

- The thermal properties were discussed in brief without clarification of the effect of modification on the thermal stability of the prepared membrane you should report the initial decomposition temp, the temperature at which the maximum decomposition rate occurs (Tmax) and the final residue at 800 oC. In addition, you should add DTGA to show the stages of decomposition if present

The following paper can help you  

Radiation crosslinking of acrylic rubber/styrene butadiene rubber blends containing polyfunctional monomers - ScienceDirect

Mechanical, acoustical and flammability properties of SBR and SBR-PU foam layered structure - ScienceDirect

Effect of iron poly(acrylic acid-co-acrylamide) and melamine polyphosphate on the flammability properties of linear low-density polyethylene | SpringerLink 

Reviewer 2 Report

Self cleaning materials are popular topic in creation of dirt repellent clothing and anti-icing surfaces for planes. However, regarding the present article I am more interested in hydrophobicity part, since in my experience mixing of hydrophobic material and amphiphilic material may result in a very interesting dependence of properties on the pretreatment and operation conditions. That being said, can the authors please clarify if the contact angles of created membranes were measured in their dry or water equilibrated state? Because the hydrophobicity of dry and of hydrated silica would strongly differ.

I also should note that English language looks of sometimes, even for my not native speaker eyes. For example, I would start the title as "Fabrication OF superhydrophobic composite membranes <...>. See also in abstract, line 17: "The result shows, when the content of SiO2 nanoparticles was increased to 1.5 wt% <...>" - such syntax asks for the starting value, before the increase, to be listed". Those were two examples, I ask the authors to please check the entire manuscript. Of you do, please also check the subscripts in SiO2, it is missing sometimes.

Otherwise I must note that the article has very good graphic material which was very helpful for understanding and very pleasing to look at.

Reviewer 3 Report

Referee's comments

to the paper entitled "Fabrication superhydrophobic composite membranes with hierarchical and porous structures for oil/water separation"

by Chunling Zhang et al.

The paper is devoted to very interesting problem, which is important gtom the practical point of view. Hydrophobic membrane has been developed for oil removal from water. In order to reach hydrophobicity,  inorganic constituent has been inserted into polymer matrix. As a result, honeycomb-like pores was formed. The rejection of oil is 99.4% is reached for the modified PVDF membrane.

As for my opinion, the paper could be published after some additions and corrections.

First of all, I would recommend to remove " hierarchical" from the title and tect, since this assumption is not confirmed. The resolution of images is changed in 20 times (Fig. 10), it is not enough to suggest hierarchical structure (the 1st, 2nd, 3rd levels etc.). I would recommend "Fabrication of superhydrophobic composite membranes with honeycomb porous structure for oil/water separation". That's why I recommend major revision.

Introduction. Indicate, that oxides of multivalent elements are used for hydrophilization of membranes due to high content water, which is located at the surface of inorganic particles. Hydrophilicity protects membranes against fouling. Please refer to  https://doi.org/10.1016/j.desal.2013.10.024, https://doi.org/10.2298/APT1647153M, https://doi.org/10.1016/j.memsci.2008.12.014

However, your approach allows one to provide gydrophobicity.

Pores looks too large in order to provide rejection. If it is possible, the authors are invited to give pore size distribution  (according to the method of nitrogen adsorption-desorption). But this is recommended, when the authors have a possibility.

Fig. 4. What is a reason of a maximum for the DPS-1.5 membrane?

Subsection 3.2. Please change a title.

Round 2

Reviewer 1 Report

no comment

Reviewer 2 Report

First of all I would like to thank the authors for fast and extensive edit. I would point out some little problems residing or reappearing (such as the instruction on how to work with the template not being deleted, and I think I see "drpped" in line 147). My biggest problem is the English phrasing at some points, such as "Maintained the system for 5–10 min to ensure completed separation, used two barrels to collect oil and water. The calculation formula " at lines 158-159, where the first sentence looks like it misses the subject and suddenly starts with a verb and in the beginning of the second sentence I would say "The equation" instead of "The calculation formula". But again, not a native speaker here, I might be wrong myself. Would the authors please double check just to be safe?

I would also recommend the authors to measure the contact angle of the swollen membranes at some time in the future.

Best regards.

Reviewer 3 Report

As for my opinion, I would recommend to publish the paper.  For a future, I advice to perform porosity measurements.
